# USP7 Inhibitors in Cancer Immunotherapy: Current Status and Perspective

**DOI:** 10.3390/cancers14225539

**Published:** 2022-11-10

**Authors:** Georgiy Korenev, Sergey Yakukhnov, Anastasia Druk, Anastasia Golovina, Vitaly Chasov, Regina Mirgayazova, Roman Ivanov, Emil Bulatov

**Affiliations:** 1Scientific Center for Translational Medicine, Sirius University, 354349 Sirius, Russia; 2Institute of Fundamental Medicine and Biology, Kazan Federal University, 420008 Kazan, Russia

**Keywords:** ubiquitin-specific protease 7, cancer, immunotherapy, small-molecule inhibitors

## Abstract

**Simple Summary:**

Elevated levels of ubiquitin-specific protease 7 (USP7) are associated with poor survival in many cancers. This protein impairs the balance of various cellular proteins and regulates the anti-tumor immune response. Its inhibition affects regulatory T cells and tumor-associated macrophages and decreases levels of various oncogenic markers. Thus, the application of USP7 inhibitors alone and in combination with cancer immunotherapeutics is highly promising. We review the potential benefits of the application of USP7 inhibitors for cancer immunotherapy and their interplay with other cancer therapeutics.

**Abstract:**

Ubiquitin-specific protease 7 (USP7) regulates the stability of a plethora of intracellular proteins involved in the suppression of anti-tumor immune responses and its overexpression is associated with poor survival in many cancers. USP7 impairs the balance of the p53/MDM2 axis resulting in the proteasomal degradation of the p53 tumor suppressor, a process that can be reversed by small-molecule inhibitors of USP7. USP7 was shown to regulate the anti-tumor immune responses in several cases. Its inhibition impedes the function of regulatory T cells, promotes polarization of tumor-associated macrophages, and reduces programmed death-ligand 1 (PD-L1) expression in tumor cells. The efficacy of small-molecule USP7 inhibitors was demonstrated in vivo. The synergistic effect of combining USP7 inhibition with cancer immunotherapy is a promising therapeutic approach, though its clinical efficacy is yet to be proven. In this review, we focus on the recent developments in understanding the intrinsic role of USP7, its interplay with other molecular pathways, and the therapeutic potential of targeting USP7 functions.

## 1. Introduction

Ubiquitination is a crucial posttranslational modification (PTM) that supports intracellular signal transduction and controls the localization, stability, and activity of numerous proteins [1,2,3]. A certain class of proteases known as deubiquitinases (DUBs), or deubiquitinating enzymes can reverse the ubiquitination process. The ubiquitin-specific protease family (USP), represented by over 50 enzymes, is the largest and the most diverse among all DUBs. USPs have a significant impact on the regulation of numerous biological processes and cellular pathways, including the cell cycle, chromatin remodeling, deoxyribonucleic acid (DNA) damage repair, ribonucleic acid (RNA) maturation, and protein synthesis [4]. A particular member of this family, USP7, also known as herpesvirus-associated protease (HAUSP), has drawn significant attention from the research community in recent years due to its important role in the regulation of anti-tumor responses [5]. Its significance in carcinogenic pathways has led to substantial interest in the discovery of potent USP7 inhibitors with drug-like properties and well-defined mechanisms of action [6]. This is further reinforced by the involvement of USP7 in the modulation of the immune response, considering the growing role of immunotherapy in cancer treatment.

Immunotherapy has become a preferred treatment modality for many cancers. It employs a wide range of molecular mechanisms to normalize impaired antitumor immunity and enable the patient’s own immune system to eliminate tumor cells [7,8,9]. Although immunotherapy can also result in notable side effects, in general, it is better tolerated than traditional chemotherapy. In many cases, the positive outcome persists for up to several years after the immunotherapy treatment [10,11,12,13]. Immunotherapy was shown to be effective in the treatment of multiple solid tumors such as melanoma, non-small cell lung cancer, renal cell carcinoma, bladder cancer, etc. [14,15,16]. However, immunotherapy has not become a panacea because not all cancers respond to the treatment equally well. Treatment failure caused by the development of resistance is observed in up to 60% of patients with the initial response to immunotherapy [17]. Even cutting-edge therapeutics such as checkpoint inhibitors often demonstrate limited efficacy, which is attributed to the existence of multiple pathways that allow tumor cells to evade immune surveillance. Therefore, searching for novel therapeutic modalities that can enhance the efficacy of immunotherapy treatment is of utmost importance.

USP7 represents a highly lucrative target since it modulates the degradation of multiple proteins involved in the regulation of the antitumor immune response. Moreover, USP7 inhibition directly impacts the critical p53/MDM2 axis that controls the programmed death of tumor cells. Here, we summarize the recent advances in our understanding of how USP7 inhibitors influence the anti-tumor immune response in vivo.

## 2. DUBs: Cellular Functions and Their Role in the UPS System

Ubiquitination is a process in which one or more small 8.6 kDa ubiquitin (Ub) molecules are attached to protein substrates by a covalent isopeptide bond. Among other outcomes, this leads to the degradation of the protein by means of 26S proteasome, a key component of the UPS system that conducts cleavage of ubiquitinated target protein into smaller peptide chains [18,19,20,21]. Such processes of protein recycling are essential for the elimination of incorrectly folded or aggregated proteins as well as proteins that are no longer needed for cell functioning [22]. Pathological alterations in the UPS pathway can cause malignant cell transformations [23,24].

DUBs are the enzymes that mediate the removal of covalently attached Ub moieties from the substrate proteins by hydrolyzing the isopeptide bond between them. DUBs do not only counterbalance ubiquitin signaling by reversing the Ub attachment, but also facilitate the recycling of Ub molecules and regulate their intracellular levels [25,26]. DUBs often include several protein-interaction domains for binding specific target proteins; they prefer to cleave specific ubiquitin branches, such as K48- or K63-linked chains (Lys48 or Lys63-linked Ub, respectively). DUBs have various expressions and the subcellular localization in vivo indicating the possibility of substrate specificity [27].

DUBs activity is tightly regulated by substrate-induced conformational change, adaptor proteins binding, proteolysis, or by PTMs [28]. Such regulation is necessary to control cell biology and physiology, whereas defects in DUBs turnover may cause neurodegenerative diseases, inflammatory and autoimmune disorders, infections, and malignancies. It has been reported that DUBs not only regulate the levels and activation of various oncogenes or tumor suppressor proteins, and support tumor development by controlling key epigenetic changes, but they also are involved in immune system regulation [29].

Over 100 DUBs are known to be encoded in the human genome [30,31]. Most of them are cysteine proteases, which can be classified into five major families according to their sequence and structural similarities: ubiquitin-specific proteases (USPs), ubiquitin carboxyl-terminal hydrolases (UCHs), ovarian tumor proteases (OTUs), Machado–Joseph (Josephin) domain-containing (MJD) proteases, and motif interacting with Ub-containing novel DUB family proteases (MINDYs). Among them, ubiquitin-specific proteases form the largest DUB family that has a common conservative domain architecture, which includes so-called Finger, Palm, and Thumb motifs. Palm and Thumb moieties form an active site of the enzyme, while Finger coordinates the linker cleavage process [32].

USP7 is one of the most well-studied enzymes among the whole DUB class [29,33]. It is a 135 kDa cysteine protease that consists of seven structural domains: N-terminal Tumor necrosis factor Receptor-Associated Factor (TRAF)-like domain, catalytic core domain, and five C-terminal ubiquitin-like domains (UBL1-5) [34,35,36,37]. The catalytic core domain (residues 208–560), the largest domain of USP7, is located between TRAF-like and UBL domains, linking Palm and Thumb motifs of the enzyme. The highly conservative catalytic core is present in all USP family members, and its main function is to cleave an isopeptide bond between Ub and the substrate protein [38,39]. In USP7 the core is formed by Cys223, His464, and Asp481 amino acid residues that cooperatively participate in a substrate deubiquitination process [32].

Within the whole USP enzyme family, only USP7 contains the unique TRAF-like domain (residues 53–205) that is necessary for substrate recognition. The C-terminal ubiquitin-like domain (residues 564–1102) contains five ubiquitin-like motifs. The end of the C-terminal tail is formed by a disordered conservative peptide (residues 1080–1102), which performs regulatory functions [40]. The catalytically active spatial arrangement of these domains takes place upon interaction with a ubiquitinated substrate through conformational change [32,40,41].

## 3. USP7 as a Regulator of Anti-Tumor Immune Response

Considerable progress has been made over the past several years in understanding the role of USP7 in modulating the immune response in cancer patients. However, the precise mechanism by which USP7 affects antitumor immunity remains to be elucidated.

Increased levels of USP7 were shown to enhance tumor progression by modulating the immunosuppressive functions of Forkhead box P3 (Foxp3)+ T-regulatory (Treg) cells [42,43,44,45,46]. Accumulation of Foxp3+ Tregs within tumor tissue and/or draining lymph nodes was confirmed as the negative prognostic factor in many solid tumors, including lung and colon cancer. Thus, Treg depletion or functional suppression could be a promising clinical strategy for cancer immunotherapy [47,48].

USP7 deubiquitinates and stabilizes histone acetyltransferase tat-Interactive protein (Tip60) and transcriptional regulator Foxp3, which are essential for the survival of Treg cells [49]. This results in the Tip60-mediated formation of Foxp3 dimers that bind to DNA and activate the transcription of anti-inflammatory cytotoxic T-lymphocyte-associated protein 4 gene (CTLA4) and interleukin (IL-10) genes while inhibiting the expression of pro-inflammatory IL-2 and interferon-gamma (IFN-γ) genes. This suggests that therapeutic USP7 inhibition may reduce the immunosuppressive functions of Tregs within the tumor microenvironment and thereby facilitate the elimination of tumor cells [49,50].

USP7 was identified as an important and highly expressed gene in the homeostasis of M2, but not M1 macrophages. Specific suppression of USP7 by small interfering ribonucleic acid (siRNA) or USP7 inhibitors was found to result in phenotypic and functional alterations in M2 macrophages, which promoted the cluster of differentiation CD8+ T cell proliferation in vitro [51]. When evaluated in mice with Lewis lung carcinoma, USP7 inhibitor P5091 slowed tumor growth and promoted the infiltration of M1 macrophages and IFN-γ-expressing CD8+ T cells. Depletion of tumor-associated macrophages (TAMs) diminished these therapeutic effects. It has been demonstrated that USP7 inhibition mediates macrophage reprogramming by activating the p38 mitogen-activated protein kinase (MAPK) pathway. The administration of USP7 inhibitors increased the expression of programmed cell death PD-L1 in tumors, which, in combination with the inhibition of programmed cell death protein 1 (PD-1), elicited a potent antitumor response. The results of this study show that targeting USP7 can modulate the immunosuppressive effects within the tumor microenvironment and improve the efficiency of lung cancer immunotherapy.

USP7 is overexpressed in some cancer types, e.g., gastric tumor cells, and its levels often correlate with the expression of PD-L1 immune checkpoint inhibitor, which suggests that USP7 might facilitate the stabilization of PD-L1 [51].

## 4. USP7 and the p53/MDM2 Axis

In addition to its immunomodulatory functions, USP7 regulates the activity of crucial transcription factors, such as p53, enabling synergistic targeting of different molecular mechanisms. The well-known tumor suppressor protein p53 is essential for cell cycle regulation, DNA repair, cell differentiation, and apoptosis. E3 ubiquitin ligase MDM2 is a negative regulator of p53 and primarily functions via forming a negative feedback loop with p53.

Importantly, USP7 was first discovered as a p53-interacting protein that strongly stabilized p53 by direct deubiquitination even in the presence of excess MDM2 and induced p53-dependent tumor cell growth suppression and apoptosis [52]. USP7-mediated deubiquitination plays a primary role in the maintenance of the p53/MDM2 axis via the stabilization of both p53 and MDM2. However, USP7 overexpression leads to more profound deubiquitination of MDM2 compared to that of p53, which results in p53 degradation and progression of tumorigenic processes [53]. Therefore, the inhibition of USP7 to enhance p53 stabilization is one of the promising strategies for anticancer therapy with high efficiency demonstrated both in vitro and in vivo [54].

USP7 overexpression stabilizes p53 even in the presence of excess MDM2, whereas USP7 downregulation renders endogenous p53 unstable [55]. USP7 also binds and stabilizes MDM2 under normal conditions, resulting in p53 turnover in an MDM2-dependent manner. When a decrease in USP7 levels leads to MDM2 degradation, the available MDM2 levels become insufficient for ubiquitination, which stabilizes p53.

The USP7/p53/MDM2 axis remains one of the cornerstones of the USP7 pathway in the nucleus, and fresh studies continue to illustrate how USP7 promotes p53-dependent apoptosis in disease. For example, in esophageal cancer USP7 inhibition increased phorbol-12-myristate-13-acetate-induced protein 1 (NOXA) expression, which then resulted in p53-dependent apoptosis [56].

The regulatory link between USP7 activity and p53 status is considered an important area of research for the development of novel antitumor therapeutics. USP7 inhibition was shown to induce tumor cell death caused by the accumulation of DNA damage in chronic lymphocytic leukemia (CLL) [57]. Based on mouse xenograft models, CLL was found to be associated with upregulated USP7 that contributed to aberrant homologous recombination repair, and sensitized p53-deficient, chemotherapy-resistant CLL to clinical doses of chemotherapeutic agents. Targeting USP7 with specific inhibitors may thus have therapeutic promise even in tumors with faulty p53 or resistant to ibrutinib chemotherapy.

Another study found that the USP7-specific submicromolar reversible inhibitor, HBX41108, was able to stabilize p53 and p21, as well as block the proliferation of HCT116 colorectal carcinoma cells [58]. HBX41108 stabilized p53, stimulated transcription of p53 target genes without causing genotoxic stress, and suppressed cancer cell growth at a level comparable to the effect of RNA interference-mediated USP7 silencing.

Andrew Turnbull et al. reported two compounds, FT671 and FT827, which inhibited USP7 in vitro and in vivo in human cells with high affinity and specificity [59]. In contrast to other USP deubiquitinases, co-crystal structures suggested that both compounds target a dynamic region near the catalytic core of the protein. Consistent with the expected USP7/target interaction mode in cells, FT671 destabilized USP7 substrates including MDM2, increased p53 levels, enhanced transcription of p53 target genes, and suppressed tumor development in mice.

As mentioned earlier, USP7 inhibitors increase PD-L1 levels, which renders tumor cells more amenable to anti-PD-1 therapy. The combination of USP7 inhibition with anti-MDM2 therapy and PD-1 blockade may appear highly effective in enhancing cancer immunotherapy. The role of p53 in immune modulation and combination therapy with PD-1 blockade was investigated using APG-115, a small molecule MDM2 antagonist in clinical development as a pharmacological p53 activator [60]. The results indicate that p53 activation mediated by APG-115 promoted antitumor immunity within the tumor microenvironment (TME) irrespective of the TP53 mutation status. This effect was dependent on the activation of p53 in immune cells with wild-type p53 residing in TME. A Phase 1b clinical trial (NCT03611868) evaluating the combination of APG-115 and pembrolizumab is ongoing for solid tumor patients including those with mutant p53 tumors.

## 5. USP7 as a Therapeutic Target

USP7 engages in numerous signaling pathways and regulates the stability of many proteins engaged in mitosis, apoptosis, DNA replication and reparation, cell cycle, epigenetic regulation, and immune response [61]. Over the past decade, more than 70 binding partners and potential substrates of USP7 have been discovered, among them tumor suppressors, oncoproteins, viral proteins, transcription factors, chromatin-associated proteins, cell cycle checkpoint proteins, and epigenetic modulators [62]. This list is regularly updated with new USP7 substrates. During the past two years, several new proteins were confirmed as USP7 substrates, including Protein phosphatase 2A (PP2A) [63], MAX dimerization protein (MGA) [64], Pleckstrin homology domain interacting protein (PHIP) [64], X-linked inhibitor of apoptosis protein (XIAP) [65], ATP binding cassette subfamily B member 1 (ABCB1) [66], and RAF proto-oncogene serine/threonine-protein kinase (Raf-1) [67]. Emerging evidence supporting the key role of USP7 in multiple diseases leads to its recognition as a promising target for drug development [68].

As described earlier, normal USP7 functions are necessary for the maintenance of cellular homeostasis, while its aberrant expression may lead to the development of cancer. USP7 overexpression in tumors is often associated with metastasis, drug resistance, and reduced patient survival [69,70].

USP7 contributes to the development of drug resistance by stabilizing proteins involved in certain signaling pathways. Therefore, USP7 inhibition in combination with other therapeutic approaches not only has a synergistic impact that improves the treatment efficacy, but also helps overcome tumor chemoresistance [70,71]. Cartel et al. demonstrated the role of the USP7/Checkpoint kinase 1 (CHK1) axis in the formation of resistance in acute myeloid leukemia [71]. Elevated levels of CHK1 protein, generated as a result of USP7-mediated deubiquitination, facilitate the reset of the replication fork during DNA replication and the adaptation of cells to cytarabine-induced DNA damage. The combination of cytarabine with the USP7 inhibitor P22077 has a synergistic impact on promoting anti-leukemic activity and can help overcome chemoresistance in cancer cells.

USP7 inhibition may be important in overcoming therapeutic resistance. Yu et al. demonstrated that combining the USP7i compound with trastuzumab slows down tumor growth in patient-derived xenografts (PDX) from a patient with a positive human epidermal growth factor receptor 2 (HER2+). This justifies the use of combination therapy in cancers with high levels of HER2 signaling, such as HER2+ breast cancer that do not respond to anti-HER2 therapy [72]. Furthermore, in chemotherapy-sensitive PDX models of small-cell lung cancer, Grunblatt et al. observed that overexpression of either MYCN or MYCL conferred chemoresistance to cisplatin–etoposide. USP7i resensitized chemoresistant mice in vivo to chemotherapy, either alone or in combination with cisplatin–etoposide [73].

However, the development of candidate molecules was slow largely due to difficulties related to suboptimal activity and selectivity [74]. Since Colland et al. reported in 2009 the pioneering series of USP7 inhibitors [58], numerous types of ligands have been designed to target either the active site of the protein, the nearby catalytic cleft, or the allosteric palm site of the catalytic domain in both covalent and non-covalent mode (Figure 1) [3,5,53,75,76]. Although the influence of these compounds on various pathways in the development of cancer malignancies has been investigated thoroughly, their ability to modulate anti-tumor immune response was evaluated only for a small number of previously reported USP7 inhibitors (Figure 2; Table 1).

Among them, by far the most extensively studied one is the electron-deficient thiophene derivative P5091 [77,78]. Others include structurally related sulfide P217564 [49,79], 4-hydroxypiperidine derivative Almac4 [80], covalently binding 9-chloro-1,2,3,4-tetrahydroacridine derivative HBX19818 [81], and tricyclic 4-ethylpyridine derived ligand GNE-6776 [82,83] (Table 1).

All covalent USP7 inhibitors target Cys223, which forms the catalytic triad together with His464 and Asp481, and partially occupies the nearby so-called “catalytic cleft” (Figure 1, purple) by binding to USP7 in the catalytic center (Figure 1, green). Various structural types of non-covalent inhibitors were developed; however, only two major modes of binding appear to be possible: binding inside the catalytic cleft typically forming interaction with Val296 (e.g., Almac4) (Figure 1, purple); and binding to the allosteric site (e.g., GNE-6776) (Figure 1, blue).

Dai et al. investigated the possibility of reprogramming TAMs to treat lung adenocarcinoma [51]. Depending on the microenvironmental cues, the macrophages can differentiate into two major phenotypes: tumor-suppressing M1 and tumor-promoting M2 phenotype [84,85]. To explore whether USP7 inhibition leads to the selective elimination of M2 macrophages, three USP7 inhibitors P5091, HBX19818, and GNE-6776 were assessed. These compounds did not exhibit any notable cytotoxic effects (at concentrations up to 10 μM). USP7 inhibition significantly lowered the levels of the M2-associated marker CD206 but did not affect the M1-associated marker CD86. Altogether this led to the conclusion that USP7 inhibition suppresses M2 macrophages. Furthermore, P5091 was used for the treatment of mice bearing a subcutaneous Lewis tumor. Its administration (40 mg/kg) caused a reduction in tumor growth by 73%. The mechanism of P5091-mediated anti-tumor effects was shown to be related to the activation of the p38 MAPK signaling pathway. A detailed examination showed that the proportion of M1 macrophages and the M1/M2 ratio increased significantly (by factors of 2.1 and 5.3, respectively). CD4+ T cell count was decreased, while an increase in the number of CD8+ cytotoxic T lymphocytes was detected. Levels of certain cytokines in the tumor microenvironment were also affected: tumor necrosis factor-alpha (TNF-α), IFN-γ, IL-2, and IL-5 were elevated, while IL-6 was decreased. USP7 inhibition contributed to TAMs polarization into proinflammatory M1 macrophages and promoted the local anti-tumor immune response in TME in vivo, while also inducing systemic adaptive anti-tumor immunity. Finally, it is noteworthy that P5091-mediated USP7 inhibition resulted in increased PD-L1 expression in tumor cells. Thus, the combination therapy based on USP7 inhibitors and anti-PD-1 antibodies could provide a synergistic impact on the efficacy of this approach.

The application of P5091 for the treatment of colorectal cancer was investigated using a mouse CT26 xenograft model [15]. The study showed that the administration of USP7 inhibitors could potentiate the efficacy of tumor vaccine therapy. Simultaneous treatment with an adenovirus-based vaccine and USP7 inhibitor demonstrated superior results compared to each agent separately. The tumor growth was reduced significantly, and the effect of a 10 mg/kg dose of P5091 was close to that of anti-PD-1 monoclonal antibodies (mAb) (5 mg/kg). P5091 decreased the levels of anti-inflammatory cytokine IL-10 and increased proinflammatory IFN-γ and TNF-α in tumor tissue and TME. Cytotoxic effects of CD4+ and CD8+ T cells against tumor cells were enhanced. Furthermore, the Foxp3 levels in CD4+ T cells were downregulated, indicating potential suppression of Tregs. 

Based on the chemical structure of P5091 second-generation dual USP7/47 inhibitors were developed exhibiting improved affinity and DUB selectivity profile [86]. The most investigated among them was P217564, which possesses the α,β-unsaturated ester moiety allowing USP7 inhibition in an irreversible manner [49,79]. Both P5091 and P217564 were evaluated for their ability to block the immunosuppressive functions of Treg cells. The compounds demonstrated a long-lasting inhibitory effect on murine Treg cells in vitro at 10 μM. USP7 inhibition downregulated both Foxp3 and its positive regulator Tip60. Moreover, the inhibitors impaired the suppressive impact of Tregs on the proliferation of co-transferred effector T cells (Teffs) and obstructed the Treg-dependent permanent allograft survival. Together, these results show that USP7 inhibition by P5091 and P217564 selectively interferes with the immunosuppressive functions of regulatory T cells, while preserving other important host T cell antitumor responses. Additionally, the influence of USP7 inhibition on tumor growth was assessed using a mouse model of E7+ TC1 lung adenocarcinoma for which the growth is Treg-dependent. For this purpose, P217564 was utilized at 1 and 6 mg/kg daily doses. Tumor growth was significantly abrogated in the case of wild-type (WT), but not immunodeficient, mice. The detailed analysis showed that at higher P217564 doses the intratumoral accumulation of Tregs was reduced; in both cases, tumor infiltration by CD8+ T cells was increased. Further, USP7 inhibitors were used to treat AE.17 mesothelioma, which is another Treg-dependent cell line. As for the TC1 tumor, the administration of USP7 inhibitors impaired tumor growth in WT mice but had no significant effect on immunodeficient mice. The negligible effect of USP7 inhibition on the growth of TC1 and AE.17 tumors in immunodeficient mice may be attributed to the lesser importance of both USP7 itself and its substrates p53 and MDM2 in these cell lines. The desired tumor suppression could be achieved only by means of immune response enhancement. This effect may be expressed in contrasting ways for different cell lines, such as MM.1S, which is hypersensitive to USP7 inhibitors. The administration of USP7 inhibitors was also found to potentiate the efficacy of adenovirus-based tumor vaccine therapy [49]. The combined treatment hindered the growth of TC1 tumor cells more efficiently than either component alone. The number of CD8+ T lymphocytes and the release of proinflammatory IFN-γ was increased significantly, while the accumulation of Foxp3+ Treg cells within the tumor was reduced. In addition, USP7 inhibition improved the efficacy of anti-PD-1 mAb therapy in mice bearing the TC1 lung tumor.

Another USP7 inhibitor Almac4 was shown to decrease PD-L1 levels in several tumor cell lines in a dose-dependent manner [87]. A similar effect was found for P5091; its application decreased membrane levels of PD-L1 in tumor cells. Importantly, inhibiting USP7 not only sensitized tumor cells to T cell-mediated cytotoxicity by reducing cell surface PD-L1 levels and attenuating its interaction with PD-1, but it also decreased their proliferation by stabilizing p53 in vitro and in vivo. The latter effect was associated with the modulation of USP7 downstream targets, including p53, MDM2, and p21 proteins.

The synergistic effect of RRx-001 and P5091 was demonstrated in studies by Das et al. where multiple myeloma (MM) cells have been demonstrated to benefit from increased levels of DNA (cytosine-5)-methyltransferase 1 (DNMT1), the primary enzyme that promotes DNA methylation. USP7 enhanced DNMT1 activity, while USP7-siRNA inhibited DNMT1 functions and reduced the survival of MM cells [88].

According to Du et al. increased expression of DNMT1 correlates with high expression of USP7 in human colon cancer cells. USP7 inhibition reduced the activity of DNMT1, which had a detrimental impact on cancer cell survival. In both tissue culture and tumor xenograft models, USP7 knockdown improved the susceptibility of colon cancer cells to histone deacetylase 1 (HDAC) inhibitors. HDAC1 and USP7 maintained DNMT1, whereas Tip60 acetyltransferase destabilized it, resulting in ubiquitination by E3 ligase UHRF1 and DNMT1 proteasomal degradation. These findings suggest that a combination of HDAC1 and USP7 inhibitors might be effective in cancer treatment [89].

Lin et al. found that USP7 is a viable therapeutic target for treating triple-negative breast cancer (TNBC) chemoresistance. USP7 overexpression boosted TNBC chemoresistance, but USP7 knockdown substantially increased the chemo-sensitivity of chemoresistant TNBC. In chemoresistant TNBC, the GNE-6776 USP7 inhibitor successfully caused apoptosis and inhibited metastasis. The authors demonstrated that USP7 is a specific deubiquitinating enzyme for ABCB1, which plays a crucial role in drug resistance development. USP7 interacted with ABCB1 directly and controlled its stability, resulting in enhanced chemoresistance [66].

## 6. Conclusions

The concept of targeting proteins by modulating their half-life is an emerging strategy in drug discovery. The field of the ubiquitin–proteasome system is on the continuous rise with more therapeutically relevant targets being discovered and more drugs entering clinical trials. Both ubiquitination and deubiquitination are highly dynamic processes that are required to maintain cellular homeostasis in health and disease. Abnormal expression of DUBs can lead to cancer, viral infections, immunological disorders, and other diseases. USP7, one of the most important DUBs, stabilizes many proteins crucial to tumor cell viability, and therefore it is gaining attention as a potential target for cancer treatment. 

In this review, we briefly covered the status of USP7-targeted drug development with the focus on cancer immunotherapy and combination with other therapeutic approaches. This included information on the USP7 structure and mechanism of action, the regulatory role in anti-tumor immune response and interplay with the p53/MDM2 axis, and the discovery of inhibitors and prospective clinical applications.

Non-clinical studies of USP7 inhibitors revealed Treg modulation as the potential mechanism of action in immunotherapeutic applications. USP7 inhibition stabilizes Foxp3, reduces Treg immunosuppression, and enables immune-mediated tumor suppression. Tregs contribution to immune tolerance is well known, and anti-CTLA-4 monoclonal antibodies were used to target them in cancer patients with some success [90]. In animal models, USP7 inhibition effectively reduced the immunosuppressive function of regulatory T cells; although, further clinical validation of these findings is necessary.

Another immune evasion mechanism exploited by many tumors and targeted by USP7 inhibitors is mediated by TAMs. TAMs are also known to promote angiogenesis and metastasis in many cancers [91]. USP7 inhibition leads to TAM polarization towards pro-inflammatory M1 macrophages, potentiating anti-tumor immune responses. The diversity and plasticity of TAMs obstruct the clinical application of therapeutic approaches that selectively target mononuclear phagocytes. The complex activity of USP7 inhibitors in multiple immune cell subsets may help overcome this challenge.

USP7 inhibition leads to the modulation of multiple immune-oncology targets, offering a promising opportunity to improve the efficacy of cancer immunotherapy. The efficacy of PD-1/PD-L1 checkpoint inhibitors is limited due to multiple additional mechanisms promoting tumor immune tolerance. Thus, the development of combinatorial immunotherapy approaches has been a priority in the last decade. However, incremental gains achieved with the addition of any targeted immunotherapy to PD-1/PD-L1 inhibitors remain relatively small. The use of agents that can block multiple immune tolerance mechanisms is therefore preferable. Growing evidence suggests that combining USP7 inhibitors with immunotherapies such as anti-PD1 or anti-CTLA4 antibodies, and cancer vaccines has the potential to significantly increase antitumor immune responses. However, the efficacy of USP7 inhibitors has yet to be shown in a cancer patient population carefully selected with the use of genetic and immunological markers that reflect the complex biological role of USP7.

USP7 inhibitors increase p53 levels, induce cell cycle arrest, and ultimately cause cell death in cellular and animal models. The potential synergy of immune modulation and restoring p53 function via USP7 inhibition is of particular interest. The p53/MDM2 axis remains an attractive target for drug developers, despite multiple failed attempts to prove the clinical efficacy of this approach. Both USP7 and MDM2 inhibition approaches have demonstrated impressive results by restoring p53 in cancer cells in vitro and in vivo and reducing chemotherapy-induced damage. Patients with MDM2 overexpression and immunosuppressive tumor microenvironment may have the largest benefit from the use of USP7 inhibitors.

The key hurdle that obstructs the progress toward wider clinical application of small molecule USP7 inhibitors is related to the complexity of DUB’s biological function [53]. Different DUBs may have opposite effects on crucial immune response mechanisms. Therefore, the selectivity of USP7 inhibitors should be considered early in their development. A recently reported series of molecules with an excellent potency and selectivity profile [92,93,94,95], as well as anti-USP7 proteolysis targeting chimera (PROTAC) compounds U7D-1 [96] and CST967 [97] represent an effort to expand a range of novel approaches for USP7 inhibition.

The large number of USP7 targets inevitably leads to potential on-target side effects of USP7 inhibitors. Therefore, this class of compounds requires thorough toxicological testing; clinical studies of USP7 inhibitors should be conducted in patients with a higher benefit/risk ratio. Patients with p53 mutations resistant to multiple previous lines of therapy, including immunotherapy, would be a suitable population for early-phase clinical trials of USP7 inhibitors.

Therefore, further studies should focus on identifying synergistic DUB inhibition mechanisms and developing combinatorial therapeutic approaches for cancer therapy. The development of novel bioassays and disease models will also facilitate this progress.

## Figures and Tables

**Figure 1 cancers-14-05539-f001:**
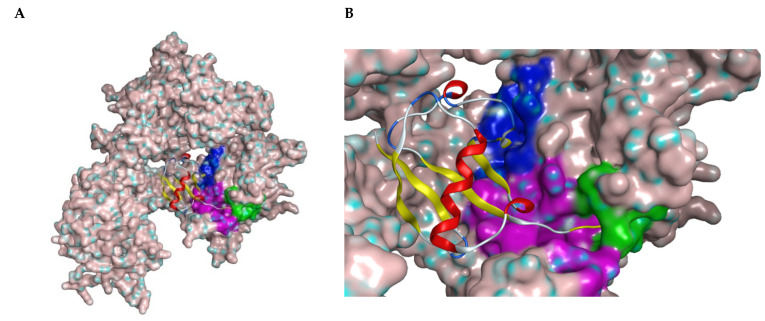
(**A**) Structure of USP7 catalytic domain in complex with ubiquitin (PDB ID: 1NBF). The molecular surface was added. The ribbon model represents ubiquitin molecule. (**B**) Closer look of the binding region of USP7 inhibitors. The green surface depicts the catalytic center; the purple surface depicts the cleft between the Thumb and Palm domains; the blue surface depicts the allosteric binding site.

**Figure 2 cancers-14-05539-f002:**
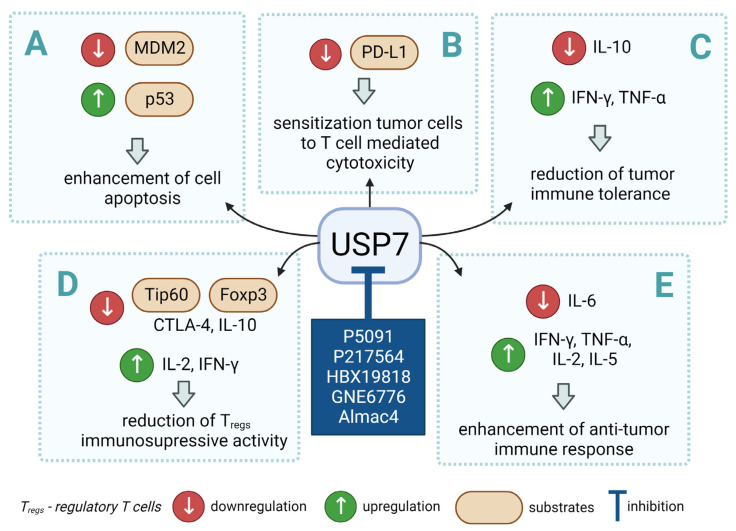
Small molecule inhibition of USP7 and its effect on anti-tumor immune response: (**A**) Initiation of p53-dependent apoptosis in tumor cells via p53/MDM2 pathway. (**B**) Decreasing PD-L1 levels sensitize tumor cells to T cell-mediated cytotoxicity by reducing PD-L1 interaction with PD-1. (**C**) Overcoming the immune tolerance of tumor via lowering IL-10 levels. (**D**) Reducing immunosuppressive functions of Tregs via destabilization of Tip60 and Foxp3. (**E**) Promotion of local anti-tumor immune response in tumor microenvironment via polarization of macrophage phenotype from anti-inflammatory M2 to proinflammatory M1.

**Table 1 cancers-14-05539-t001:** Biological properties of the selected USP7 inhibitors with demonstrated effect on anti-tumor immune response.

USP7 Inhibitor	Mode of Action	Cancer Type In Vitro and/or In Vivo Studies
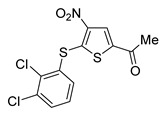 P5091	Covalent Non-selective (dual USP7/47) IC50 (USP7) 4.2 μM IC50 (USP47) 4.3 μM	Multiple myeloma MM.1S, MM.1R, U266, KMS12PE, etc. Colorectal cancer HCT116 Prostate cancer PC3, LNCaP Ovarian cancer SKOV3, OVCAR-8, HeyA8 Urothelial bladder cancer J82, T24, KU-19-19 Esophageal squamous cell carcinoma HN5, HN30, UM-SCC-1, UM-SCC-25, etc. Chronic lympholytic leukemia CCRF-CEM Medulloblastoma D823, Daoy
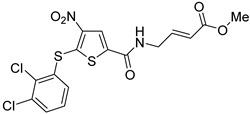 P217564	Covalent Non-selective (dual USP7/47) IC50 (USP7) 0.48 μM IC50 (USP47) 0.66 μM	Colorectal cancer HCT116 Prostate cancer PC3
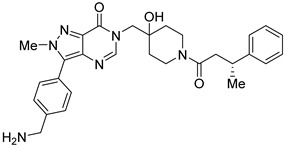 Almac4	Non-covalent Selective IC50 (USP7) 0.0015 μM IC50 (USP47) > 50 μM	Colon cancer HCT116 Breast cancer LNCaP Osteosarcoma SJSA-1 Prostate cancer MCF7
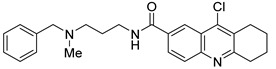 HBX19818	Covalent Selective IC50 (USP7) 28.1 μM IC50 (USP47) NR ^1^	Colon cancer HCT116 Osteosarcoma U2OS, SJSA-1
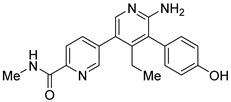 GNE-6776	Non-covalent Selective IC50 (USP7) 1.3 μM IC50 (USP47) > 200 μM	Colon cancer HCT116 Osteosarcoma SJSA-1 Acute myeloid leukemia EOL-1 Breast cancer MCF7

^1^ NR—Not reported.

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
