# Peer review of "USP7 Inhibitors in Cancer Immunotherapy: Current Status and Perspective"

_cancers, 2022, doi:10.3390/cancers14225539_

Round 1

Reviewer 1 Report

Comments to the Author

In this article, the structural characteristics of USP7 and the current treatment status of USP7 inhibitors were discussed in details by authors. Although this study needs some revision, it does provide novel information on the USP7 as novel drug target in cancer therapy. However, the manuscript should be improved by addressing the following concerns before publication. My opinion is major revision.

The major concerns:

1. The abbreviation has been given, the full name may not be used.

2. Notably, USP7 may further play a key role in overcoming therapeutic resistance. For example, a report showed that the combination of USP7i and trastuzumab can synergistically inhibit tumor growth. This provides a rationale for combination therapy in HER2+ breast cancers that do not respond to HER2 therapy, as well as in other cancers where HER2 signaling is upregulated. ( Cancer Res. 79 (13), 2980. doi:10.1158/1538-7445.AM2019-2980)

3. The authors can explore some difficulties and challenges in the treatment of USP7 inhibitors and add prospects for future clinical application

4. When listing USP7 inhibitors, especially covalent binding drugs, the authors should also give the binding domain of the drug.

5. The authors did provide a detailed interpretation of the current status of USP7 inhibitors for immunotherapy, which has been reviewed recently by other authors, e.g., Front Chem. 2022 Sep 15; 10:105727. The author needs to add some more novel ideas to supplement the review. For example, the interaction between USP7 and DNMTs and the prospect of the combination application of related drugs and USP7 inhibitors. ( J. Cell. Biochem. 112, 439–444)

6. In addition, the authors can provide a brief list of proteins known to be regulated by USP7.

Author Response

Dear Sir or Madam,

You can find the full text of our reply in the attached file.

Sincerely yours,

Georgiy Korenev

Reviewer 2 Report

Overall, this is a clear and well-written review. The table and figures are well presented. My only criticism is that the articles cited in the review are relatively old, and some latest articles about USP7 are not cited. The regulation of p53 by USP7 has been already widely recognized, and some viewpoints are not new enough. This review would be greatly improved if some novel roles of USP7 could be summarized by citing new references.

I only have a few comments on the current version:

1)      Line115: “Increased levels of USP7 were shown to enhance tumor progression by modulating the immunosuppressive functions of Forkhead box P3 (Foxp3+) T-regulatory (Treg) cells.” Reference should be added. “Forkhead box P3 (Foxp3+)” should be corrected as “Forkhead box P3 (Foxp3)+”.

2)Tumor-associated macrophages (TAMs) abbreviation first appears in Line 134. So, tumor associated macro-phages (TAMs) in line 255 and 347 should be the abbreviation only.

Author Response

Dear Sir or Madam

You can find the full text of our reply in the attached file

Sincerely yours, 

Georgiy Korenev

Round 2

Reviewer 1 Report

  • There are no defects in the manuscript and the requirements for publication have been met after revision.